# Real-time monitoring of multimode squeezing

Mahmoud Kalash 1,2 ✉, Aditya Sudharsanam 1,2, M. H. M. Passos 1,3, Valentina Parigi 4 & Maria Chekhova 1,2 ✉

Multimode squeezed light is a key resource for high-dimensional quantum technologies, enhancing metrological sensitivity, boosting communication security, and enabling parallel processing in computation. Its practical potential, however, remains constrained by the inherent single-mode operation of homodyne detection, necessitating post-processing for multimode characterization. Here, we overcome this long-standing challenge by employing multimode optical parametric amplification, enabling loss-tolerant direct detection of squeezing in each mode, which in turn permits mode sorting after amplification. As a result, we demonstrate, for the first time to the best of our knowledge, the real-time monitoring of multimode squeezing. With a spatial light modulator sorting the modes, we simultaneously measure squeezing in nine spatial modes co-propagating within one beam. Although mode sorting and filtering reduce the detection efficiency to less than 0.3%, we observe high-purity squeezing of up to − 7.9 ± 0.6 dB − to the best of our knowledge, the highest squeezing recorded for pulsed light. Furthermore, we demonstrate real-time, loss-tolerant characterization of continuous-variable entanglement and extend it to the detection of cluster states. Similar methods can be applied in the frequency domain, facilitating a crucial capability for scalable quantum technologies.

Optical quantum information science promises significant advancements in photonic technologies and heralds a new era of applications. Among quantum photonic resources, squeezed light stands out as the most fundamental and relatively the easiest to generate, while offering a wide range of capabilities[1,2]. Even more important is multimode squeezed light, as it extends these capabilities to high-dimensional scenarios[3,4], making them viable for real-life applications. These include sub-shot-noise sensing and imaging[5–9], secure communication through continuous-variable (CV) quantum key distribution[10], boson sampling[11,12], and quantum computing via cluster states[13,14].

Despite these advantages, high-dimensional scenarios have been handled inefficiently so far, limiting the full potential of multimode squeezed states. This is mainly because homodyne detection (HD)[15], the standard CV technique for characterizing quantum states, is inherently limited to single-mode operation. Specifically, HD employs

a local oscillator (LO), which defines the detected mode, restricting the simultaneous retrieval of information across multiple modes. Typically, spectral and spatial modes are addressed one by one by shaping the LO[4,16,17]. Efforts to circumvent this limitation have taken several approaches. In boson sampling experiments, multiple separate squeezed light sources were used along with a corresponding number of homodyne detectors[11] – a resource-demanding approach. Alternatively, temporal modes have been employed along with delay loops and time de-multiplexers for the creation of temporal cluster states[13,18]. Some works used array detectors to address multiple modes simultaneously[19–22], but the information was retrieved only through post-processing. As these methods are all based on HD, they also share additional limitations. The results are highly susceptible to detection inefficiency, an issue particularly critical for array detectors. Moreover, the detection optical bandwidth is limited to that of the LO, posing

[1]Max Planck Institute for the Science of Light, Erlangen, Germany. [2]Friedrich-Alexander Universität Erlangen-Nürnberg, Erlangen, Germany. [3]Centro Brasileiro de Pesquisas Físicas, Rio de Janeiro, RJ, Brazil. [4]Laboratoire Kastler Brossel, Sorbonne Université, ENS-Université PSL, CNRS, Collège de France, Paris, France. ✉e-mail: mahmoud.kalash@mpl.mpg.de; maria.chekhova@mpl.mpg.de

challenges when dealing with states more broadband than the LO. Finally, the sampling rate is limited by the electronic bandwidth of HD.

Optical parametric amplification overcomes the technical limitations imposed by HD[23–28]. By sufficiently amplifying a certain quadrature $x^{(\phi)}$ while de-amplifying the conjugate quadrature, a phase-sensitive optical parametric amplifier (OPA) maps the variance of $x^{(\phi)}$ onto the output mean intensity $I^{(\phi)}$,

$$I^{(\phi)} \propto e^{2G}\mathrm{Var}(x^{(\phi)}), \tag{1}$$

where $G$ is the amplification gain. To infer the scaling factor between the output intensity and $\mathrm{Var}(x^{(\phi)})$, one measures the OPA output for the case of vacuum at its input (calibration procedure). Given sufficiently strong amplification, the detection becomes tolerant to practically any post-amplification loss as the latter simply scales down the output intensity. In addition, OPAs are typically broadband, which makes them compatible with wide-spectrum sources. Indeed, since squeezed light is often generated using OPAs[1] and shares their spectral properties, it is naturally well-suited to be detected by OPAs. Finally, with direct detection used after the OPA, the electronic bandwidth of HD is no longer a limiting factor.

Most importantly, OPAs are usually spatially and spectrally multimode[29–32]. A multimode optical parametric amplifier (MOPA) thus has the unique capability of amplifying multiple modes at once[28,33]. Its modal content, determined by the nonlinear phase matching and the pump spectrum, can be engineered[34,35] to match any set of input modes, including complex modes such as orbital angular momentum ones[30]. Then, the output intensity distribution $I(\mathbf{r})$ can be represented as a sum of contributions from different modes,

$$I(\mathbf{r}) = I_0 \sum_{m,n} \lambda_{mn}|u_{mn}(\mathbf{r})|^2, \tag{2}$$

where $I_0$ is the integral output intensity, $\lambda_{mn}$ is the weight of spatial mode $(m, n)$, with $\sum_{m,n}\lambda_{mn} = 1$, and $u_{mn}(\mathbf{r})$, normalized as $\int|u_{mn}(\mathbf{r})|^2 d\mathbf{r} = 1$, are the shapes of the modes. The integral intensity of each mode,

$$I_{mn} = I_0\lambda_{mn}, \tag{3}$$

will then define the quadrature squeezing in this mode according to Eq. (1). Therefore, the quadrature squeezing and anti-squeezing of all modes can be retrieved by finding the weights $\lambda_{mn}$[28].

Here, we employ MOPA to monitor, for the first time to the best of our knowledge, multimode squeezed light in real time. By spatially expanding the pump, we engineer the shapes of MOPA spatial modes to better match those of the input quantum state. Tolerance of MOPA-based detection to loss allows us to sort the output spatial modes – a procedure that is inevitably lossy. In particular, we measure high-purity squeezing simultaneously in nine spatial co-propagating modes within one beam. This further enables the real-time characterization of CV entanglement of 36 mode pairs. Moreover, we consider possible cluster states formed by various mode combinations and evaluate their quality. The simultaneous access to multiple squeezed modes unlocks the full potential of high-dimensional quantum applications, especially ones requiring dynamic control and feedback, including quantum metrology, communication, and continuous-variable quantum computation.

## Results

Figure 1a presents the generalized setup for the detection of multimode squeezing, here applied to spatial modes. The multimode squeezed light to be monitored is injected into a MOPA with modes tailored to match the input, which is achieved by shaping the pump. In the simplest case, depending on the pump phase, either the squeezed or anti-squeezed quadratures are amplified for all modes.

Alternatively, by engineering the pump field, different quadratures can be addressed for different modes, enabling the detection of more complex states. Afterwards, a mode sorter, in this case a spatial light modulator (SLM), multiplexes different modes or mode superpositions, which are then monitored through an array detector (see Supplementary for details). Importantly, after the amplification, the detection becomes tolerant to losses, in particular those accompanying sorting and noise. Finally, if the MOPA amplifies quadrature $x^{(\phi)}$ of mode $(m, n)$, the degree of squeezing or anti-squeezing for this quadrature is found as $10\mathrm{Log}\left(\frac{I_{mn}^{(\phi)}}{I_{mn}^{\mathrm{vac}}}\right)$, where $I_{mn}^{(\phi)}$ is the mean integral intensity of the mode after amplification, and $I_{mn}^{\mathrm{vac}}$ is the mean integral intensity of the corresponding amplified vacuum mode, measured with the MOPA input blocked[36–38].

As an example, we examine a spatially multimode squeezed vacuum state occupying a set of Hermite-Gaussian (HG) modes that co-propagate within a single beam. Figure 1b shows the real-time traces of the quadrature variances of five exemplary modes ($\mathrm{HG}_{00,11,02,21,22}$), all acquired simultaneously after the MOPA stage. Because in our setup the squeezed modes are imaged on the MOPA modes (see Methods and Supplementary Information), we detect squeezed or anti-squeezed quadratures simultaneously for all spatial modes. As the pump phase $\phi$ is scanned with time, the quadrature that is amplified goes from the squeezed to the anti-squeezed and so on.

### Mode matching

We generate spatially multimodal squeezed vacuum using type-I collinear frequency-degenerate parametric down-conversion (PDC) in a 3 mm nonlinear crystal pumped by picosecond pulses. Such a source can generate squeezed vacuum with various spatial mode contents, depending on the pump spatial profile. In our proof-of-principle experiment, we chose the simplest Gaussian pump. The resulting set of spatial modes is close to Hermite-Gaussian beams[39–41], with the size determined by the crystal length and the pump beam waist. Other modal contents can be generated by shaping the pump differently[42]. By focusing the pump into a waist of $97 \pm 2\,\mu m$ inside the crystal, we obtain an effective number of 53 spatial modes, the strongest (Gaussian) mode having a waist of $23\,\mu m$. We set the squeezing parameter for the collinear emission to $G_{sq} = 1.05 \pm 0.2$, corresponding to approximately 9 dB of squeezing. This value, measured without mode selection and thus reflecting the collective contribution of all spatial modes, is used as a reference to set the squeezing level of the individual modes. Finally, we image the squeezed vacuum modes on the MOPA. Through alignment, we achieve a uniform intensity distribution, thus making sure that all input modes are amplified or deamplified simultaneously (Methods). See the Supplementary Information for more details on the modal content of the squeezed vacuum.

For the MOPA, we employ the same crystal but stronger pumping (gain $G = 4.4 \pm 0.3$ for collinear emission) to make Eq. (1) applicable and achieve an acceptable signal-to-noise ratio while amplifying the squeezed quadrature[27]. We characterize the MOPA modal content experimentally, by analyzing the spatial intensity correlations at its output[41] (Methods). We aim at monitoring squeezing in nine amplifier detection channels, which correspond to MOPA's $\mathrm{HG}_{00,01,10,11,02,20,12,21,22}$ modes. Ideally, those modes should match an equivalent set of input squeezed modes, which is generally not the case. In the experiment, because the MOPA has a higher gain than the squeezer, the angular sizes of MOPA spatial modes are larger than those of the squeezed vacuum modes[37]. Therefore, directly imaging the squeezed modes into the MOPA leads to mode mismatch. This problem can be fixed by increasing the MOPA pump waist. In more general cases of arbitrary mode structures, tailoring the pump profile would be required.

We increase the MOPA pump waist to $145\,\mu m$, 1.5 times larger than the squeezer's pump. This adjustment partially compensates for the

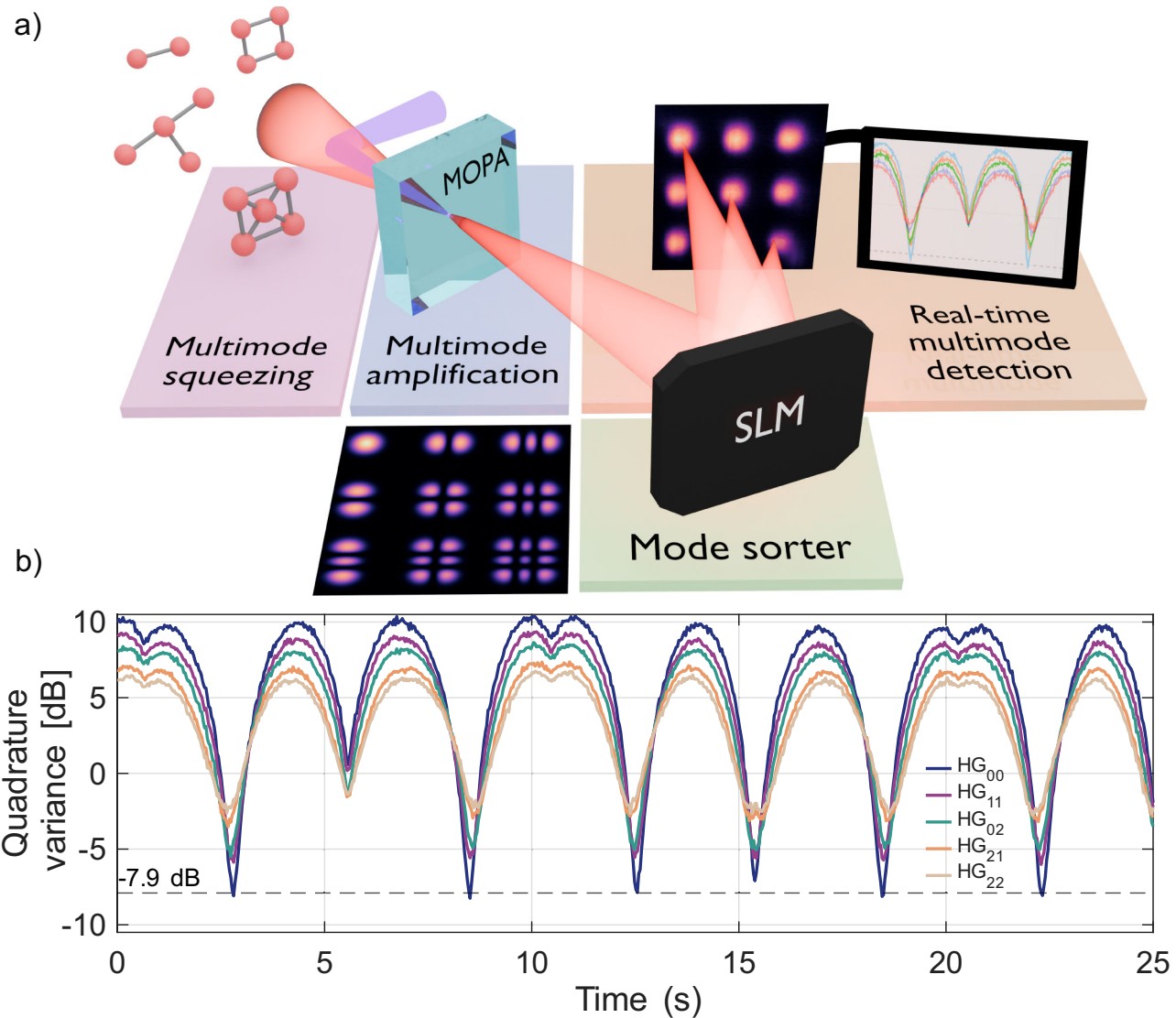

**Fig. 1 | Real-time monitoring of multimode squeezing. a** Generalized setup for real-time detection of spatially multimode squeezed light: after its amplification with a multimode optical parametric amplifier (MOPA), modes are sorted using a spatial light modulator (SLM). By measuring their intensities with a camera, we infer the variances of the input mode quadratures. The choice of the quadrature is determined by the pump phase. **b** Traces of quadrature variances for five squeezed Hermite-Gauss modes $HG_{mn}$, obtained in real time by scanning the pump phase.

angular broadening caused by the higher gain, thus improving the mode matching. Figure 2a shows the calculated overlap $|\kappa_{m,n,k,l}|^2$ between nine modes $HG_{mn}$ of the squeezer and $HG_{kl}$ of the MOPA (see Methods). The red bars show the improvement in mode matching resulting from the optimized pump beam waist in the amplifier. The achieved overlap is more than 85% for $HG_{00,10,01}$ and more than 70% for $HG_{11,12,20,02,21}$. Mode $HG_{22}$ shows the most pronounced improvement, with its overlap increasing from 38% to 64% as a result of pump shaping. The residual overlap leads to the coupling between the targeted input modes and other amplifier modes, described by the overlap matrix. Accordingly, the observed MOPA signals reflect the squeezing redistributed among its modes. This redistribution can be accounted for, and the squeezing of each input mode can be retrieved, if all MOPA modes are accessed (Methods). However, the correction is small (see Supplementary Information, Section 2.3).

### Real-time measurement of squeezing over multiple modes

The SLM sorts the amplifier modes (see Supplementary Information) by converting each mode into a Gaussian-like spot at a certain location on a camera (Fig. 1a), enabling access to several modes simultaneously.

The contributions $I_{mn}$ of these modes to the output intensity are measured by integrating over square areas at the centers of the spots. From these values, we infer the quadrature variances and measure them while scanning the MOPA pump phase, as shown in Fig. 1b.

Figure 2b shows the squeezing and anti-squeezing measured for all nine modes (points). Theoretically calculated values are plotted by solid lines, the shaded area showing the uncertainty caused by the gain measurement of the squeezer. The calculation takes into account the losses between the squeezer and the amplifier, assumed to be 10%. The highest retrieved degrees of squeezing and anti-squeezing are $-7.9 \pm 0.6$ dB and $10 \pm 0.5$ dB, respectively, and are observed for the fundamental mode. For higher-order modes, they are gradually reducing, which matches with the expectations for our source: the higher the mode order $m + n$, the lower the squeezing. The deviation from the theory for higher-order modes can be explained by the difficulty in adjusting their phases through alignment. Indeed, our alignment procedure relies on the uniformity of the intensity distribution, and higher-order modes affect it much less than the lower-order ones.

Finally, Fig. 2c shows the purity of the measured squeezed modes. Despite the huge detection losses (more than 99% mainly due to the

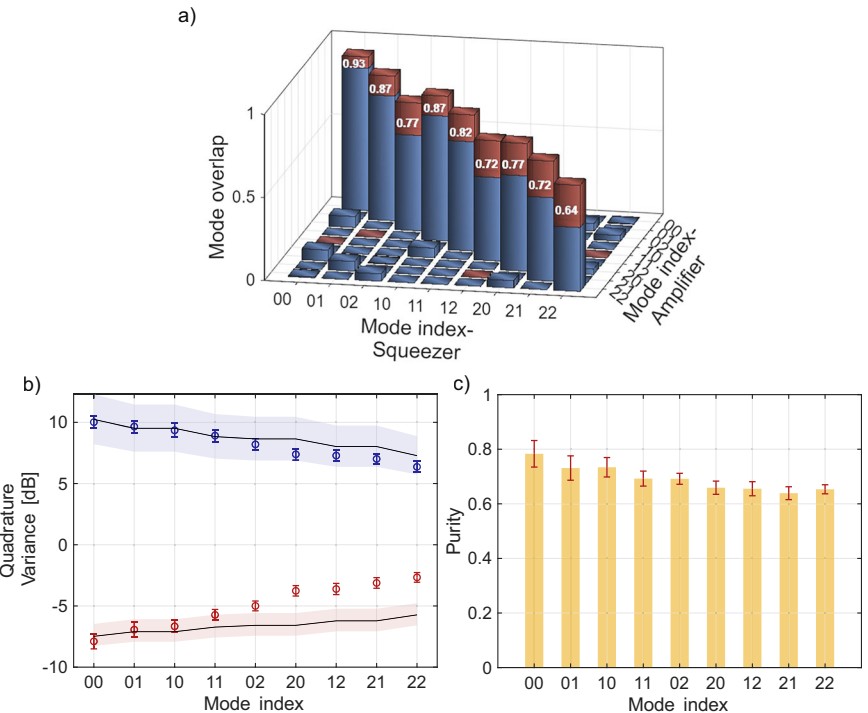

**Fig. 2 | Characterization of the first nine spatial modes. a** Overlap of the MOPA modes with the modes of the squeezer without (blue bars) and with (red bars) the MOPA pump waist optimization. **b** Anti-squeezing (blue) and squeezing (red) measured for the nine strongest spatial modes. Solid lines show the result of the theoretical calculation, accounting for 10% losses. The shaded areas correspond to the uncertainty in the gain measurement. **c** Quantum state purities obtained in experiment for all nine modes. Everywhere, error bars correspond to standard deviation.

spatial mode sorting (see the Supplementary Information), the purity is between 63% and 78%. Such a high purity is due to the parametric amplification of the state before detection.

## Efficient detection of cluster states

Having simultaneous access to individual modes, we can now build and characterize cluster states, a fundamental resource in continuous-variable quantum information[14]. These states can be generated in superpositions of squeezed-vacuum modes, forming optical nodes that are interconnected and share entanglement. A key quantity used to characterize the correlations between these cluster nodes is the nullifier $\delta$. It is evaluated based on the quadratures $X_i$ and $P_i$ of the cluster nodes as $\delta_i = \frac{P_i - \sum_j^{h_i} X_j}{\sqrt{1+h_i}}$, where $h_i$ denotes the number of neighboring nodes. Sub-shot-noise squeezing of $\delta_i$ indicates quantum correlations between the nodes of the cluster. Rather than measuring the quadratures of individual nodes separately, it is sufficient to measure the nullifier modes[4]. In HD, this is done by shaping the local oscillator into a superposition of the squeezing modes to match the modes of the nullifiers. A full characterization of a cluster state thus requires addressing all nullifiers individually, a task that becomes increasingly demanding as the cluster size grows[13,43], requiring significant time and effort. To overcome this challenge, we propose MOPA as an efficient approach for cluster state detection. By shaping the modes of the amplifier to match those of the nullifiers, all cluster links can be characterized and monitored in real time.

## Real-time detection of two-node clusters

The simultaneous measurement of squeezing for multiple modes immediately provides a tool to characterize the entanglement of their superpositions. The first example is Laguerre-Gauss modes $LG_0^1$ and $i LG_0^{-1}$, which are superpositions of squeezed modes $HG_{10}$ and $HG_{01}$: $LG_0^1 = (HG_{01} + i HG_{10})/\sqrt{2}$ and $i LG_0^{-1} = (i HG_{01} + HG_{10})/\sqrt{2}$. These modes

form the simplest cluster state: a two-node cluster, featuring EPR-like entanglement. In this case, each nullifier variance is given by the quadrature variance of one contributing squeezed mode, and so can be fully characterized by simply monitoring these modes, $HG_{01}$ and $HG_{10}$ (see Methods). The orange and blue dashed lines in Fig. 3a show the variances of these nullifiers (i), as well as the nullifier variances for similar two-node cluster states whose nodes are superpositions of $HG_{21}$ and $HG_{02}$ (ii) and $HG_{00}$ and $HG_{22}$ (iii). The variances are normalized to the corresponding shot-noise levels. All nullifiers are measured in real time as the phase of the pump is scanned, and their variances change with a period $\Delta t$. We see that for certain phases (shaded areas), all nullifiers are squeezed well below the shot-noise level.

The difference between a two-node cluster and a CV EPR state is that for the latter, the $x$-quadrature difference and the $p$-quadrature sum for two modes are squeezed simultaneously[44], while for the former, squeezed is a nullifier that is a combination of $p$ and $x$ quadratures for different modes. A witness of entanglement for modes 1 and 2 of a cluster is the sub-shot-noise value of $W \equiv \text{Var}(\delta_1) + \text{Var}(\delta_2)$[45,46] (0 dB corresponds to both $\delta 1$ and $\delta 2$ being in vacuum states). Solid bordeaux lines in Fig. 3a show this witness for three EPR-like cluster states, whose amplitudes and phases, together with the shapes of the nullifiers, are presented in the right panel of Fig. 3a. The shaded areas highlight regions where the entanglement is simultaneously verified for the three addressed states. In addition, owing to the real-time access to nine modes, we simultaneously monitored the remaining 33 two-node cluster states, demonstrating the parallel detection capability of our approach. The witnesses for all 36 two-node clusters are shown in Fig. 3b. Almost all the monitored pairs beat the -3 dB threshold (solid line), making them suitable for various quantum applications.

## Larger cluster states

Consider now clusters containing more than two nodes. Figure 4a shows some examples: a three-node, a four-node, and a five-node cluster formed by superpositions of HG modes. As an example, Fig. 4b,

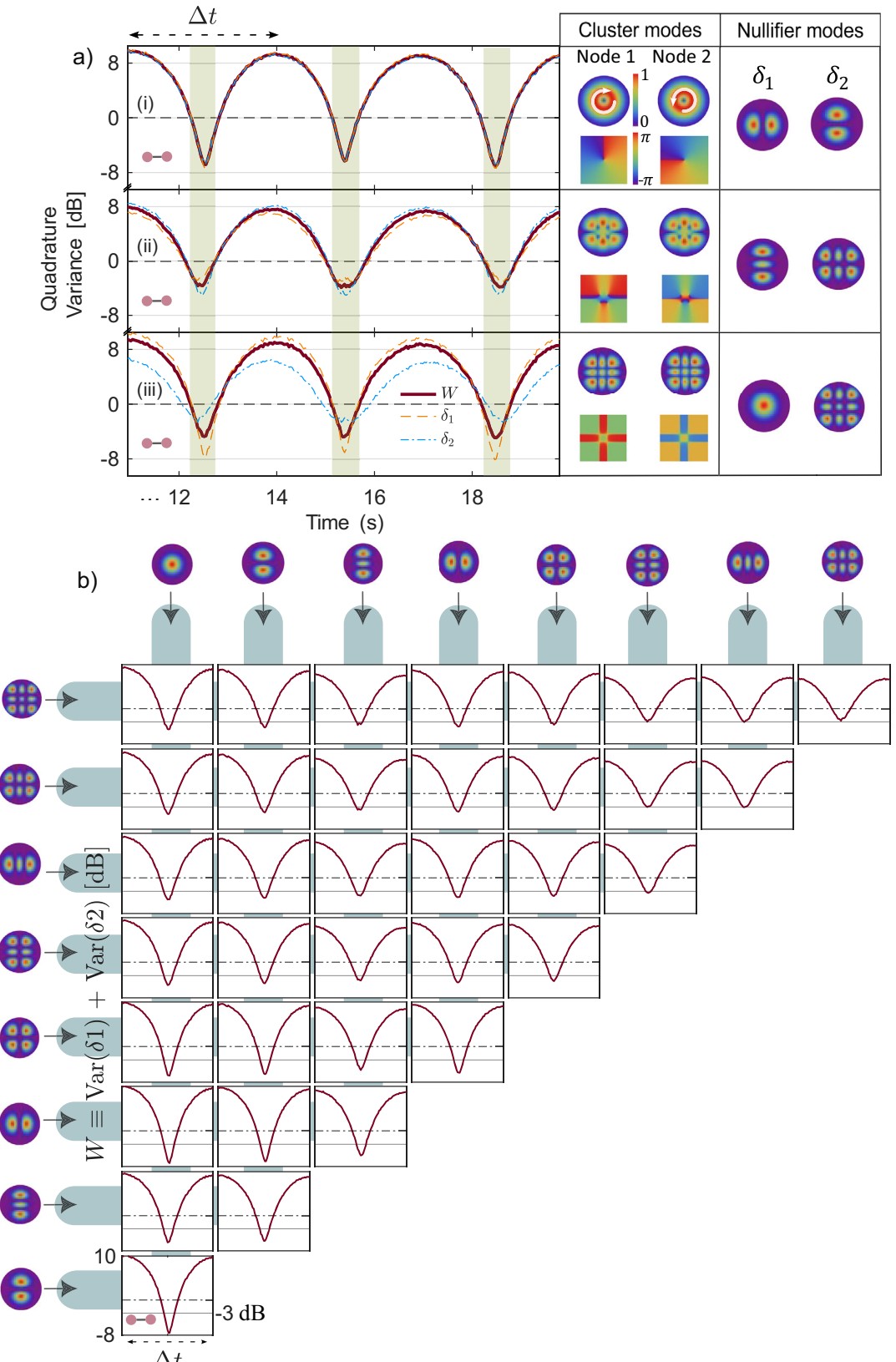

**Fig. 3 | Simultaneously monitoring two-node cluster states in real time. a** Real-time traces of the quadrature variances for nullifiers $\delta_1$ (dashed orange) and $\delta_2$ (dashed-dotted blue) of two-node cluster states, as well as the entanglement witnesses $W \equiv \mathrm{Var}(\delta_1) + \mathrm{Var}(\delta_2)$ (solid bordeaux) as the MOPA phase is scanned. The shaded areas highlight regions where the entanglement is simultaneously verified for the three addressed states. Right panel: Cluster nodes (left) and their corresponding nullifier modes (right): $HG_{01}$ and $HG_{10}$ (i); $HG_{11}$ and $HG_{00}$ (ii); $HG_{22}$ and $HG_{11}$ (iii). **b** Entanglement witnesses $W$ for all 36 pairs of entangled modes within $\Delta t$, the latter shown in panel (a). Dash-dot lines mark the 0 dB level and solid lines, the −3 dB level.

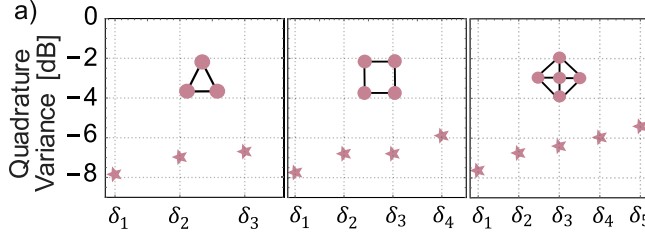

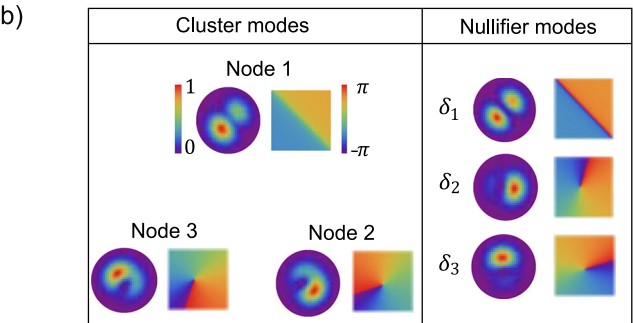

**Fig. 4 | Two-dimensional cluster states of more than two nodes. a** Nullifier variances of 3-, 4-, and 5-node clusters calculated from the measured variances of the contributing squeezed modes. Error bars (standard deviation) are within the symbols. **b** Spatial amplitude and phase profiles of node (left) and nullifier (right) modes for the three-node cluster. In this case, the involved squeezed modes are $HG_{00}$, $HG_{01}$, and $HG_{10}$.

left panel, shows the nodes of the three-node cluster. Their nullifiers (Fig. 4b, right panel) have contributions from different modes, and thus their variances cannot be measured directly without specially shaping the modes of the MOPA. However, we can calculate the squeezing of each nullifier from the measured quadrature squeezing of these contributing modes. As Fig. 4a shows, we expect rather high degrees of squeezing: from $-7.9 \pm 0.6$ dB to $-5.3 \pm 0.3$ dB (see Supplementary Information for all nullifiers variances). In our case, the contributing modes exhibit varying degrees of squeezing, leading to different levels of nullifier squeezing within a single cluster. This variation can be mitigated by appropriately selecting the contributing modes and their respective weights[4] or by tailoring the source's modal content. To characterize the nullifiers of three-node or more complex clusters, engineering of the MOPA modal structure is needed.

## Discussion

With the help of multimode optical parametric amplification, we have demonstrated real-time monitoring of squeezing for multiple spatial modes co-propagating within a single beam. After the simultaneous sufficient amplification of all modes, losses no longer affect the measurement, enabling mode de-multiplexing − an inevitably lossy procedure. In particular, in our experiment, the efficiency of mode sorting with an SLM was ≈0.5%. By sorting the modes, we were able to simultaneously access individual squeezed modes. Despite less than 0.3% detection efficiency, we observed squeezing of $-7.9 \pm 0.6$ dB with a 78% purity in the fundamental Gaussian mode, which, to the best of our knowledge, is the highest squeezing achieved for pulsed squeezed light to date. We believe that with further improvements to the setup and proper engineering of the source modal content, our multimode pulsed squeezing could reach the −10 dB threshold required for fault-tolerant quantum computation[47,48], making it a promising candidate for this important direction.

In addition to the multimode capability, our experiment highlights the natural suitability of OPA-based detection for squeezed light. Notably, at the signal wavelength of 709.33 nm, obtaining a pulsed local oscillator with matched spectral and temporal properties is technically challenging, rendering balanced homodyne detection impractical. In contrast, the OPA scheme enables such measurements by using the same source for both generation and detection, without requiring additional equipment.

Because the amplifier operates at a higher gain than the squeezer, their modes differ in size. Therefore, to optimize the measurement, we have implemented mode matching between the squeezer and the MOPA, achieving more than 70% overlap for the first eight modes by making the amplifier pump broader than the source pump. Matching for more modes can be easily provided by further expanding the pump beam for the MOPA.

While our focus has been on spatial multimode squeezing, our source also generates spectrally multimode squeezing, with many frequency modes present within each spatial mode. As in the spatial domain, spectral mode mismatch arises due to the higher gain of the amplifier, potentially degrading the retrieved squeezing per spatial mode. However, our source exhibits nearly flat squeezing across many modes, particularly within the strongest ones that dominate each spatial mode[49]. This suppresses the impact of spectral mismatch and enables accurate squeezing measurements across the full detection bandwidth, which was set to 10 nm in our experiment.

If necessary, spectral mismatch can be reduced by shaping the OPA pump spectrum, for example, through narrowband filtering, analogous to our spatial pump tailoring. Indeed, in contrast to multimode squeezing detection, this additional step becomes necessary for the tomography of a multimode quantum state, which can be readily achieved by extending our setup[27]. In that case, only a single spectral mode must be allowed for each spatial mode to ensure single-mode detection, which requires good spectral mode matching.

In turn, spectral modes can be sorted via nonlinear frequency conversion[50] or temporal interferometers[51], where the loss tolerance provided by MOPA will be highly beneficial. This is particularly relevant for integrated photonics platforms, such as those based on nonlinear optical waveguides[26,52,53], where quantum state generation and processing take place within a single spatial mode, making frequency or temporal multiplexing essential for high-dimensional operations.

The spatial modes of a paraxial beam form a rich system, where orthogonal mode bases can be chosen in infinitely many ways. For this reason, squeezing measured for individual modes in one basis (Hermite-Gauss) opens a path for constructing numerous cluster states out of modes in other bases, for instance, Laguerre-Gauss. For two-node clusters (EPR-like states), the nullifiers are directly measurable from the squeezing measured simultaneously for HG modes. In particular, we measured real-time traces of entanglement witnesses for exemplary modes $LG_0^1$ and $iLG_0^{-1}$, as well as 35 other two-mode combinations.

Here, we have demonstrated a proof-of-principle experiment in which the MOPA simultaneously amplifies either the squeezed or the anti-squeezed quadratures across multiple modes, enabling the detection of the aforementioned multimode squeezing. This allowed us to estimate the nullifier variances of exemplary three-, four-, and five-node cluster states. Furthermore, by properly engineering the amplitude and phase of the amplifier pump, different quadratures can be selectively addressed across the modes, which is essential for the real-time detection of individual cluster nodes or nullifiers. These modes are linear combinations of the original squeezed modes and generally include phase terms. In this way, MOPA enables efficient detection of larger cluster states, allowing simultaneous characterization and monitoring of all cluster links within a single measurement. We emphasize that this cannot be achieved by simply changing the output sorted modes without first modifying the MOPA modal structure.

We stress that although here we only characterized nine strongest modes, the effective number of spatial modes in our setup is about 50 and can be further increased to a few hundred by using broader pump beams. This scalability paves the way for the generation of large-scale spatially multipartite entangled states.

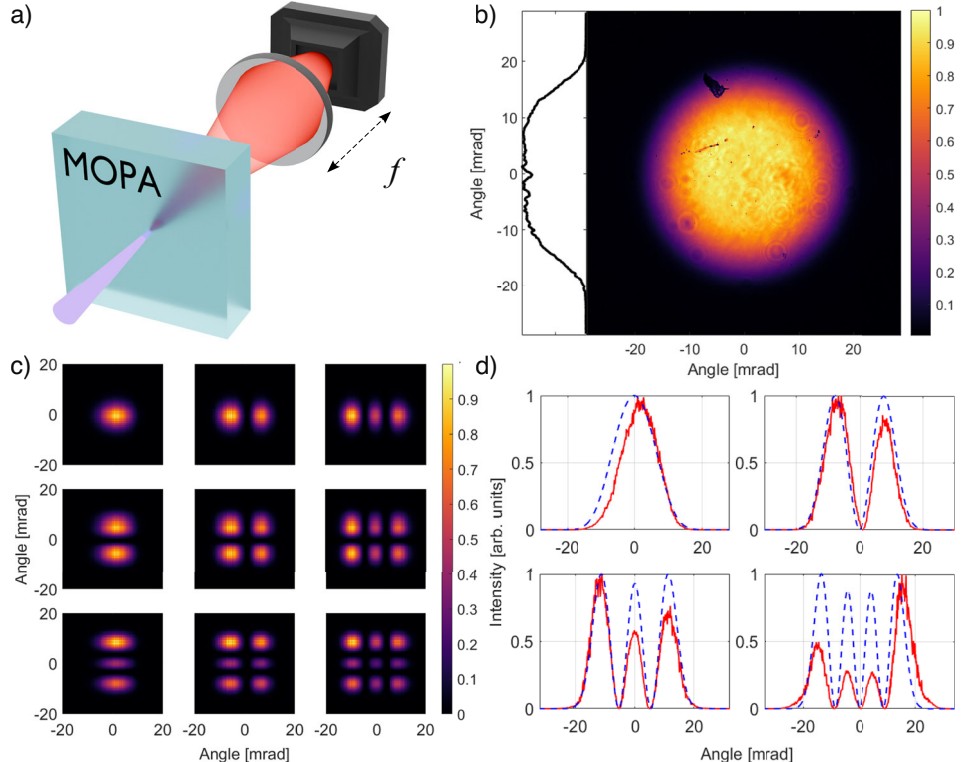

**Fig. 5 | Reconstruction of the MOPA spatial modes. a** The setup, with $f$ denoting the focal length of the lens. **b** Far-field intensity distribution $I(\theta_x, \theta_y)$. **c** Mode shapes $|u_{mn}(\theta_x, \theta_y)|^2$. **d** Calculated (dashed, blue) and measured (solid, red) one-dimensional mode shapes $|u_m(\theta_x)|^2$ for $m = 1, 2, 3, 4$.

Altogether, this work completes the set of capabilities needed for robust multimode squeezed light detection and significantly expands its applicability in emerging quantum technologies.

## Methods

### Experimental setup

We employ a wide-field SU(1,1) nonlinear interferometer in a folded scheme[24]; a single 3mm bismuth triborate (BiBO) crystal cut for type-I collinear degenerate phasematching is used for both the generation and amplification of multimode squeezed vacuum (see the Supplementary Information, Fig. S1). The pump is the third harmonic of an Nd-YAG laser with 354.67 nm wavelength, 18 ps pulse duration, 1 kHz repetition rate and a maximum pulse energy of 70 μJ. When pumped in one direction (pump 1), the BiBO crystal generates multimode squeezed vacuum. The multimode radiation is then imaged back into the crystal in a one-to-one configuration using a dichroic mirror, followed by a spherical mirror. Phase-sensitive amplification is achieved by introducing a second pump (pump 2) propagating in the same direction as the back-reflected radiation. The phase between pump 2 and the input signal is scanned using a piezoelectric actuator. Importantly, we align the spherical mirror in such a way that the far-field intensity distribution after amplification oscillates as a whole, without interference rings. Under this condition, all modes contribute to the output field with equal phases, so that in the 'dark fringe', amplified are squeezed quadratures for all modes, while in the 'bright fringe', the anti-squeezed quadratures are amplified[28]. The power of pump 1 is set at 8 mW to achieve $G_{sq} = 1.05 \pm 0.2$, theoretically corresponding to approximately 9 dB of squeezing in the collinear emission (in the absence of losses). Meanwhile, pump 2 power is adjusted to achieve an amplification gain of $G = 4.4 \pm 0.3$. This satisfies Eq. (1) with a squeezing detection accuracy of > 99% for the strongest nine modes (see Supplementary Information). Additionally, it ensures an acceptable signal-to-noise ratio when amplifying the squeezed quadrature, where the output mean intensity for each mode (m,n) follows

$\langle I_{mn}^{(\phi)} \rangle = \sinh^2(G_{mn} - G_{sq_{mn}})$, where $G_{mn}$ and $G_{sq_{mn}}$ are the gains of the amplifier and the squeezer, respectively, for this mode. Afterwards, a system of lenses magnifies the far field of the amplified radiation. An SLM (Hamamatsu LCOS-SLM X10468-06), located in the far field, sorts out the amplifier modes. Finally, an sCMOS camera is used to monitor in real time the intensities of the sorted modes.

### Calculation of the modal content

To calculate the modal content for the squeezer (where measurement of mode shapes was impossible because of the low photon flux) and the amplifier (where calculation was compared with the measurement results), we run the integro-differential equations governing high-gain collinear degenerate PDC[37,38]. Namely, we find numerically the gain-dependent functions $\eta(q, q', g)$ and $\beta(q, q', g)$ from the Bogoliubov transformations connecting the input and output annihilation operators $\hat{a}^{in}, \hat{a}^{out}$:

$$\hat{a}^{out}(q, g) = \int dq' \eta(q, q', g) \hat{a}^{in}(q') + \int dq' \beta(q, q', g) [\hat{a}^{in}(q')]^{\dagger}. \quad (4)$$

Here, $q$ is a Cartesian component of the transverse wavevector and $g$ is the squeezing parameter (amplification gain) measured in the collinear direction. Finally, by applying the joint Schmidt decomposition[38] to $\eta(q, q', g)$ and $\beta(q, q', g)$, we find the one-dimensional modal content as

$$\beta(q, q', g) = \sum_n \sqrt{\Lambda_n} u_n(q, g) \psi_n(q', g), \quad (5)$$

where $\psi_n(q, g)$ and $u_n(q, g)$ are the input and output gain-dependent modes, respectively, and $\Lambda_n = \sinh^2(g_n)$ is the mean photon number in mode $n$, which defines the squeezing (amplification gain) for this mode. To pass to 2D modes, we use the factorability of the modes in $x$ and $y$ directions: $\psi_{mn}(q_x, q_y) = \psi_m(q_x)\psi_n(q_y)$, $u_{mn}(q_x, q_y) = u_m(q_x)u_n(q_y)$.

Meanwhile, the gain of a 2D mode (m,n) is $G_{mn} = \frac{g_m g_n}{g}$. See Supplementary Material for more details.

## Experimental reconstruction of the modal content

To reconstruct the MOPA mode shapes, we block the quantum state at the input, so that the MOPA amplifies only vacuum, and examine the far-field intensity distribution of the emitted radiation (Fig. 5a). This distribution spans about 20 mrad FWHM and has a flat-top profile, typical for type-I collinear-degenerate phase matching[24] (Fig. 5b). To retrieve the modes, we acquire 1250 single-shot intensity distributions $I(\theta_x, \theta_y)$, where the angles $\theta_x, \theta_y$ are related to the transverse wavevectors $q_x, q_y$ as $\theta_{x,y} = q_{x,y}/k$ and $k$ is the full wavevector. The MOPA modes form an orthogonal set with no correlations to each other, and so they are the modes with maximal achievable squeezing. As shown in refs. 31,41, these modes for the bipartite (signal+idler) system coincide with the coherent modes of a single (signal or idler) subsystem. We distinguish signal and idler subsystems by dividing the far field into the right and left, or upper and lower, parts. The coherent modes of a single subsystem can then be reconstructed from the first-order correlation function $g^{(1)}$, which, due to the thermal statistics of a single subsystem, is related to the second-order correlation function $g^{(2)}$ via Siegert's relation $g^{(2)} = 1 + |g^{(1)}|^2$. Meanwhile, $g^{(2)}(\theta_x, \theta'_x) = 1 + C(\theta_x, \theta'_x)/(\langle I(\theta_x)\rangle \langle I(\theta'_x)\rangle)$, where $C(\theta_x, \theta'_x)$ is the intensity covariance function,

$$C(\theta_x, \theta'_x) = \langle I(\theta_x)I(\theta'_x)\rangle - \langle I(\theta_x)\rangle \langle I(\theta'_x)\rangle, \tag{6}$$

calculated from the ensemble of single-shot one-dimensional intensity distributions $I(\theta_x)$ at $\theta_y$ fixed. Using all these relations, we obtain the amplifier one-dimensional spatial modes $u_m(\theta_x)$ by applying the singular-value decomposition to the measured covariance distribution as

$$C(\theta_x, \theta'_x) \propto \left[\sum_m \lambda_m u_m(\theta_x)u_m^*(\theta'_x)\right]^2, \tag{7}$$

with $\lambda_m$ defining the weight of mode $u_m(\theta_x)$. Similarly, we obtain the one-dimensional modes for the $\theta_y$ angle. Finally, to obtain the two-dimensional modes, we make use of the factorability of the modes in the case of collinear degenerate PDC: $u_{mn}(\theta_x, \theta_y) = u_m(\theta_x)u_n(\theta_y)$. Figure 5c shows the reconstructed intensity distributions of the nine strongest modes, from HG$_{00}$ to HG$_{22}$.

The mode matching between the output modes of the squeezer $u'_{mn}(\theta_x, \theta_y)$, and the input modes of the amplifier $\psi''_{kl}(\theta_x, \theta_y)$ is evaluated by the overlap integral $|\kappa_{m,n,k,l}|^2 = |\int_x d\theta_x \int_y d\theta_y [\psi''_{kl}(\theta_x, \theta_y)]^* u'_{mn}(\theta_x, \theta_y)|^2$. As the parametric gain increases, the far-field modes broaden[37]. Meanwhile, as the pump waist broadens, the far-field modes get narrower[39]. This enables matching the modes of the squeezer and of the amplifier in our case by softer focusing the pump of the latter[28].

## Detectable squeezing per mode

Imperfect mode matching leads to coupling between the squeezed input modes and the amplifier modes. This leads to the distribution of each input squeezing value over different detection channels. If matrix $|\kappa_{m,n,k,l}|^2$ is known, and all involved amplifier channels are accessible, the squeezing in input mode $(m, n)$ can be retrieved by applying weighted intensity contributions to the output signals[28],

$$\frac{\text{Var}(x_{mn}^{(\phi)})}{\text{Var}(x_{mn}^{\text{vac}})} = \sum_{k,l} |\kappa_{m,n,k,l}|^2 \frac{I_{kl}^{(\phi)}}{I_{kl}^{\text{vac}}}, \tag{8}$$

a relation valid if $G \gg G_{sq}$[28,54]. In the experiment, we sorted only nine spatial MOPA modes, which prevented such a reconstruction. However, the effect of higher-order modes is weak (Supplementary information, Section 2.3).

## Cluster states

A squeezed vacuum consisting of $N$ Schmidt modes can be described by the quadrature vector

$$\mathbf{q^s} = \{x_1^s, x_2^s, x_3^s, \ldots, x_N^s; p_1^s, p_2^s, p_3^s, \ldots, p_N^s\}, \tag{9}$$

with $x_i^s$ ($p_i^s$) being the position- (momentum-) like quadrature of the $i$th mode. The quadratures of the n-node cluster state to be obtained from the multimode squeezed vacuum,

$$\mathbf{Q^c} = \{X_1^c, X_2^c, X_3^c, \ldots, X_n^c; P_1^c, P_2^c, P_3^c, \ldots, P_n^c\}, \tag{10}$$

are found by applying a unitary transformation $U$ to the Schmidt-mode quadratures as

$$\mathbf{Q^c} = U\mathbf{q^s} = \begin{bmatrix} a & -b \\ b & a \end{bmatrix} \mathbf{q^s}, \tag{11}$$

where

$$a = (I + A^2)^{-1/2}, \tag{12}$$

$$b = A(I + A^2)^{-1/2}, \tag{13}$$

$I$ is the identity matrix, and $A$ is the adjacency matrix of each cluster topology. For instance, the adjacency matrices corresponding to the clusters considered in Fig. 4a are

$$A_3 = \begin{bmatrix} 0 & 1 & 1 \\ 1 & 0 & 1 \\ 1 & 1 & 0 \end{bmatrix}, A_4 = \begin{bmatrix} 0 & 1 & 0 & 1 \\ 1 & 0 & 1 & 0 \\ 0 & 1 & 0 & 1 \\ 1 & 0 & 1 & 0 \end{bmatrix}, \text{and } A_5 = \begin{bmatrix} 0 & 1 & 1 & 1 & 1 \\ 1 & 0 & 1 & 0 & 1 \\ 1 & 1 & 0 & 1 & 0 \\ 1 & 0 & 1 & 0 & 1 \\ 1 & 1 & 0 & 1 & 0 \end{bmatrix}. \tag{14}$$

Finally, the normalized nullifier of node $i$ can be obtained as

$$\delta_{\mathbf{i}} = \frac{P_i^c - \sum_j^N A_{ij} X_j^c}{\sqrt{1 + h_i}}, \tag{15}$$

where $N$ is the total number of nodes and $h_i$ is the number of nodes adjacent to the $i$th node.

As an example, for a 2-node cluster state, which has the adjacency matrix

$$A_2 = \begin{bmatrix} 0 & 1 \\ 1 & 0 \end{bmatrix}, \tag{16}$$

the nodes quadratures will be written as

$$X_1^c = \frac{x_1^s - p_2^s}{\sqrt{2}}, \quad X_2^c = \frac{x_2^s - p_1^s}{\sqrt{2}}, \tag{17}$$
$$P_1^c = \frac{p_1^s + x_2^s}{\sqrt{2}}, \quad P_2^c = \frac{p_2^s + x_1^s}{\sqrt{2}}.$$

Meanwhile, their nullifiers and variances take the following form:

$$\delta_1 = \frac{P_1^c - X_2^c}{\sqrt{2}} = p_1^s, \quad \delta_2 = \frac{P_2^c - X_1^c}{\sqrt{2}} = p_2^s, \tag{18}$$
$$\Delta^2\delta_1 = \Delta^2 p_1^s, \quad \Delta^2\delta_2 = \Delta^2 p_2^s.$$

Clearly, in this case, each individual nullifier is directly related to one contributing squeezed mode. Since both squeezed modes are

inherently the modes of the amplifier, we can monitor the nullifiers of this cluster without further adjusting our experimental settings. This situation is realized for three two-node clusters (Fig. 3), whose nodes are superpositions of modes $HG_{01}$ and $HG_{10}$ (i), $HG_{11}$ and $HG_{00}$ (ii), and $HG_{22}$ and $HG_{11}$ (iii).

However, for more complex cases, such as the 3-nodes cluster (see the Supplementary Information for the other states), calculations lead to

$$\Delta^2 \delta_1 = 0.949967 \, \Delta^2 p_1^s + 0.0250165 \, \Delta^2 p_2^s + 0.0250165 \, \Delta^2 p_3^s$$
$$= -7.85 \pm 0.58 \, \text{dB},$$

$$\Delta^2 \delta_2 = 0.0250165 \, \Delta^2 p_1^s + 0.949967 \, \Delta^2 p_2^s + 0.0250165 \, \Delta^2 p_3^s, \quad (19)$$
$$= -6.95 \pm 0.57 \, \text{dB},$$

$$\Delta^2 \delta_3 = 0.0250165 \, \Delta^2 p_1^s + 0.0250165 \, \Delta^2 p_2^s + 0.949967 \, \Delta^2 p_3^s$$
$$= -6.70 \pm 0.45 \, \text{dB}.$$

In this case, individual nullifiers have contributions from several squeezed modes. To directly measure these nullifiers, we therefore need to engineer the modes of the amplifier.

In particular, the mode sets forming the 3-node, 4-node, and 5-node clusters in Fig. 4a are $HG_{00}, HG_{01}, HG_{10}$; $HG_{00}, HG_{01}, HG_{10}, HG_{11}$; and $HG_{00}, HG_{01}, HG_{10}, HG_{11}, HG_{02}$, respectively.

## Data availability

The experimental data used in this study have been deposited at https://figshare.com/s/333b4f83a2bcfc44843a.

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

## Acknowledgements

We thank Polina Sharapova, Dennis Scharwald, and Giuseppe Patera for illuminating discussions, and Changjin Son for helping with illustrations. This project is part of the Munich Quantum Valley, which is supported by the Bavarian state government with funds from the Hightech Agenda Bavaria. M. C. acknowledges funding from the Deutsche Forschungsgemeinschaft (grant number 499995074). This research was funded within the QuantERA II Programme (project SPARQL), which has received funding from the European Union's Horizon 2020 research and innovation programme under Grant Agreement No 101017733. M. K., A. S., and M. C. are part of the Max Planck School of Photonics, supported by BMBF, Max Planck Society, and Fraunhofer Society. M. K. was also funded by the Deutsche Forschungsgemeinschaft - Project-ID 429529648 - TRR 306 QuCoLiMa ("Quantum Cooperativity of Light and Matter"). V.P. acknowledges Agence Nationale de la Recherche (OQuLus, ANR-22-PETQ-0013).

## Author contributions

M.K. and M.C. conceived the project; M.K. and A.S. built the experimental setup; M.K., A.S., and M.P. carried out the experiment under the supervision of M.C.; M.P. carried out the theoretical calculations with the guidance of V.P.; All authors wrote, reviewed, and edited the manuscript.

## Funding

## Competing interests

The authors declare no competing interests.
