## [Transparent Peer Review file · Nature Communications]

Real-time Monitoring of Multimode Squeezing

Corresponding Author: Professor Maria Chekhova

Version 1:

Reviewer comments:

Reviewer #1

(Remarks to the Author)

Optical parametric amplification (OPA) overcomes the technical limitations of balanced homodyne detection, including tolerance to practically any post-amplification loss and a broader bandwidth. In the article, the authors extend OPA to the multimode optical parametric amplifier (MOPA). They investigate how to optimize squeezing mode matching via the pump spatial profile of MOPA to enhance measurement efficiency, and demonstrate the real-time simultaneous measurement of multimode squeezed light as a proof-of-principle experiment. Specifically, they realize the simultaneous measurement of high-purity squeezing in nine co-propagating spatial modes, along with the demonstration of multimode entanglement. To my knowledge, this is the first example of using optical parametric amplification (OPA) to realize spatial multimode squeezing measurement. It provides an effective tool for conducting continuous-variable (CV) multimode quantum information research. The submitted manuscript is worthy of publication after the authors have responded to the relevant comments and suggestions. My specific comments and suggestions are as follows:

(1) In multimode squeezing measurement, the most important aspect is how to accurately measure the squeezing of each mode; therefore, it is crucial to avoid the influence of inter-mode coupling on the measurement. Regarding the amplification process of MOPA and the mode separation process by SLM, although the authors have also considered their impacts on the measurement of squeezing values, such considerations are mostly based on theoretical estimations. I think there is no direct evidence for the accuracy of the measured squeezing values. Since balanced homodyne detection can provide standard measurement results, as a new scheme, it would be advisable to conduct mutual verification with the measurement results obtained by the traditional balanced homodyne detection, so as to provide data support for the results of this paper.

(2) Since pulsed light is used as the pump, in addition to generating spatial multimode squeezing, it also produces parametric time-frequency domain multimode squeezing. May I ask whether the time-frequency domain multimode processes, especially spatio-temporal coupling, will have an impact on spatial multimode squeezing and amplification? If so, how do they affect and how can we avoid them?

(3) The text mentions "By shaping the modes of the amplifier to match those of the nullifiers, all cluster links can be characterized and monitored in real time". Could you please provide an explanation? For example, in what scenarios is it necessary to "shape the modes of the amplifier" to achieve the measurement? Is it possible to achieve this by adjusting the projection modes of the SLM without modifying the pump?

(4) In Figure 3, why is the black line W above 0 dB (SNL)? This will mislead readers into thinking that there is no entanglement.

(5) Since this is a spatial multimode amplification process, the Guoy phase will be introduced during the propagation of the multimode field, thereby changing the squeezing characteristics. Has the impact of the Guoy phase on the measurement been considered?

(6) Will adding the physical description and corresponding theory of the MOPA measurement scheme at the beginning of the article better improve the article's readability?

(7) It can be seen from the formulas in section (6.4) of the article that the imperfect mode coupling of MOPA will reduce the measured squeezing value. In the formulas, R_{mn} represents the loss caused by imperfect mode coupling and introduces vacuum noise. However, during the MOPA process, does imperfect mode coupling only introduce vacuum noise? Are there any other amplified coupling noises? And are the formulas derived strictly?

Reviewer #2

(Remarks to the Author)

This work addresses the long-standing challenge of real-time, loss-tolerant characterization of multimode squeezed light by

proposing and experimentally validating a solution based on multimode optical parametric amplifiers (MOPAs). First, spatially multimode squeezed vacuum states (with approximately 53 effective modes, and the strongest mode corresponding to ~9 dB of squeezing) are generated via type-I collinear degenerate parametric down-conversion (PDC) in a BiBO crystal. A MOPA is then constructed using the same BiBO crystal, and mode matching with the squeezer is achieved by optimizing the MOPA pump waist (145 μm), resulting in an overlap of over 70% for the first 8 modes. Leveraging the MOPA's ability to "map quadrature variances to output intensity," post-amplification detection becomes loss-tolerant. Subsequently, a spatial light modulator (SLM) sorts the amplified multimode light (with a sorting efficiency of ~0.5%), and a camera real-time reads the intensities of the 9 strongest Hermite-Gaussian (HG) modes to infer their quadrature variances. Experimental results show that despite an overall detection efficiency of less than 0.3%, the fundamental mode achieves -7.9 ± 0.6 dB of pulsed squeezing (the highest value reported to date) with a quantum state purity of 63%–78% by mean intensity detection. Additionally, the system enables real-time characterization of nullifier variances and entanglement witnesses for two-node cluster states, estimates the nullifier squeezing levels of 3-to-5-node cluster states via mode contribution analysis, and proposes extending this method to frequency modes—providing a critical characterization tool for high-dimensional quantum technologies such as quantum metrology and continuous-variable quantum computing. Before making a decision about the presented manuscript, the following concerns should be addressed.

1. The squeezing here is come from the mean intensity detection of the mode after amplification. It is totally different from the standard method Balanced Homodyne Detection (BHD). The proof of equivalence is necessary. At least, the authors should compare the squeezing measuring between two squeezing measuring methods for the same high mode once. I think the gain of MOPA is corresponding to a possible maximum squeezing limit.
2. Based on the above, the expression of "the highest recorded squeezing" is inappropriate. However, it is reasonable if the authors give the proof of equivalence.
3. Could the authors provide specific details on how the pump beam is shaped? Additionally, it is a suggestion that relevant information about pump shaping be added to Fig. S1 for clarity.
4. The modes purity is also important. Output light after filter has been reduced to 0.3%, how much squeezing of remaining light is?

Reviewer #3

(Remarks to the Author)

Reviewer #4

(Remarks to the Author)

In their work titled "Real-time Monitoring of Multimode Squeezing", the authors have demonstrated an approach to simultaneously monitor multi-mode squeezed states in real time using a combination of a multimode OPA and SLM based mode-sorter. They used their approach to also characterize cluster states. The results in this work are not surprising and is expected, thanks to the loss-tolerant detection mechanism provided by OPA. The results in this paper are comprehensive. I am willing to recommend publication of this work in Nature communications, if the authors can address my concerns, that I believe will improve the clarity of the work.

1. Can the authors add more details about the mode overlap purity that can be obtained by this approach. is it possible to attain above fault-tolerant threshold in the context of CV cluster-state based quantum computing applications.
2. What is the degradation expected with higher order modes? What is the theoretical bounds on the useful number of modes that can be addressed while still satisfying the fault-tolerant threshold given certain constraints like pump power (that can also lead to spontaneous nonlinear scattering process that can degrade the degree of squeezing), resolution of the SLM etc..
3. Can the authors comment on the scalability of the approach, especially in the context of integrated photonics platform like lithium niobate nonlinear integrated photonic circuits. How many modes can be harnessed in this platform, given the flexibility of QPM engineering, and replacing SLM with waveguide coupler based mode-sorter?
4. How does spatial multiplexing compare with regards to frequency-domain multiplexing with regards to real-time detection. Are there concerns of cross-talk in frequency domain approach, that is less severe in mode-division approach?

Version 2:

Reviewer comments:

Reviewer #1

(Remarks to the Author)

The authors have provided a comprehensive and satisfactory response to the questions I raised. Particularly noteworthy is

that the manuscript supplements more rigorous theoretical formulas, analyses, and figures, which offer strong support for the rigor of the experiments. Therefore, I recommend accepting the article for publication.

Reviewer #2

(Remarks to the Author)

We appreciate the authors for their comprehensive and detailed responses to our initial concerns, as well as the careful revisions made to the manuscript. The supplementary explanations regarding the equivalence of the OPA-based detection scheme and BHD, the clarification of pump shaping details, and the supplement of relevant data and figures have effectively addressed our doubts. The revisions have further enhanced the scientific rigor and clarity of the work. We are now satisfied with the revised version of the manuscript and fully support its publication.

Reviewer #1 (Remarks to the Author):

Optical parametric amplification (OPA) overcomes the technical limitations of balanced homodyne detection, including tolerance to practically any post-amplification loss and a broader bandwidth. In the article, the authors extend OPA to the multimode optical parametric amplifier (MOPA). They investigate how to optimize squeezing mode matching via the pump spatial profile of MOPA to enhance measurement efficiency, and demonstrate the real-time simultaneous measurement of multimode squeezed light as a proof-of-principle experiment. Specifically, they realize the simultaneous measurement of high-purity squeezing in nine co-propagating spatial modes, along with the demonstration of multimode entanglement.

To my knowledge, this is the first example of using optical parametric amplification (OPA) to realize spatial multimode squeezing measurement. It provides an effective tool for conducting continuous-variable (CV) multimode quantum information research.

We thank the reviewer for their understanding of our work. We are pleased that the novelty and significance of the proposed approach are appreciated.

The submitted manuscript is worthy of publication after the authors have responded to the relevant comments and suggestions. My specific comments and suggestions are as follows:

(1) In multimode squeezing measurement, the most important aspect is how to accurately measure the squeezing of each mode; therefore, it is crucial to avoid the influence of inter-mode coupling on the measurement. Regarding the amplification process of MOPA and the mode separation process by SLM, although the authors have also considered their impacts on the measurement of squeezing values, such considerations are mostly based on theoretical estimations. I think there is no direct evidence for the accuracy of the measured squeezing values. Since balanced homodyne detection can provide standard measurement results, as a new scheme, it would be advisable to conduct mutual verification with the measurement results obtained by the traditional balanced homodyne detection, so as to provide data support for the results of this paper.

Indeed, the reviewer is right to point out the importance of a reliable verification method such as balanced homodyne detection (BHD). However, in our case, BHD is impossible because a suitable local oscillator at the squeezing wavelength (709.33 nm) is unavailable. This limitation actually illustrates a key strength of OPA-based detection: the same source used for squeezing generation can be employed for detection, ensuring natural compatibility, without the requirement of additional sources.

Meanwhile, squeezing detection via OPA has become a reliable method and has already been implemented in several previous works [*Nat Commun* **9**, 609 (2018), *Optica* **6**, 1233-1236 (2019), *Science* **377**,1333-1337(2022), and *Opt. Express* **28**, 34916-34926 (2020)], all of them accepted in the scientific community. Importantly, the equivalence between the OPA scheme and BHD was recently tested (*Opt. Express* **28**, 34916-34926 (2020)), proving the

reliability of the method. Finally, our results reasonably match the expectations of the rigorous theoretical model of our multimode squeezer (Fig. 2(b) of main text).

As to the inter-mode coupling, and as the reviewer noted, we have considered their effect on our system, but only theoretically. Nevertheless, we stress that we apply no correction to compensate for mode mismatch, and the reported results are fully experimental. Our calculations show that inter-mode coupling reduces the value of measured squeezing, but only slightly.

We now comment on both points, the local oscillator limitation (lines 237-241) and the effect of inter-mode coupling (lines 481-488), in the revised manuscript.

(2) Since pulsed light is used as the pump, in addition to generating spatial multimode squeezing, it also produces parametric time-frequency domain multimode squeezing. May I ask whether the time-frequency domain multimode processes, especially spatio-temporal coupling, will have an impact on spatial multimode squeezing and amplification? If so, how do they affect and how can we avoid them?

The reviewer raises a relevant point. In our system, many spectral modes are indeed present within each spatial mode, and vice versa. There are two key aspects to consider:

1) The detection bandwidth we use is 10 nm, which lies within the flat-top spectral region of the OPA output. This ensures that all frequency modes within this window are amplified homogeneously, enabling broadband detection of each spatial mode.

2) The gain of the OPA ($G = 4.4$) is higher than that of the squeezer ($G_{sq} = 1.05$), which makes the amplifier spectral eigenmodes broader than the squeezed modes. In general, this could lead to spectral mode mismatch. However, our source possesses almost flat squeezing over many spectral modes (see arXiv:2508.04502 (2025)), rendering the effect of spectral mode mismatch negligible on squeezing detection.

Nevertheless, if required, the spectral-mode mismatch can be mitigated by spectrally shaping the OPA pump (e.g., simply through pump narrowband filtering), analogous to our spatial shaping approach.

We now comment on the spectral aspect of our system in the revised manuscript (lines 247-253).

3) The text mentions "By shaping the modes of the amplifier to match those of the nullifiers, all cluster links can be characterized and monitored in real time". Could you please provide an explanation? For example, in what scenarios is it necessary to "shape the modes of the amplifier" to achieve the measurement? Is it possible to achieve this by adjusting the projection modes of the SLM without modifying the pump?

Multipartite entangled states such as cluster states are conceptually generated via multimode squeezing followed by linear interferometry. The resulting modes (e.g., cluster nodes or nullifiers) are specific combinations of the original squeezed modes, as discussed in Supplementary Section 5. We note that such modal combinations also include phase terms, requiring the MOPA to simultaneously amplify different quadratures across different modes. This means that the amplifier eigenmodes must be tailored to match the specific quadrature combinations defined by the cluster state. For this reason, it is not possible to perform such a measurement merely by changing the projection modes at the output without first modifying the OPA modal content.

We now explain this in the revised manuscript (lines 276-283).

(4) In Figure 3, why is the black line W above 0 dB (SNL)? This will mislead readers into thinking that there is no entanglement.

We thank the reviewer for paying attention to this graph. In fact, there was a mistake in the normalization of the entanglement witness W . After we corrected this error, the W lines go below the shot-noise level for all three states shown in Fig. 3, as well as for the other 33 pairs of entangled modes we previously showed in the Supplementary Section 5.5. For completeness, we now add these data to the main text (Fig. 3(b)).

(5) Since this is a spatial multimode amplification process, the Guoy phase will be introduced during the propagation of the multimode field, thereby changing the squeezing characteristics. Has the impact of the Guoy phase on the measurement been considered?

We thank the reviewer for pointing out the relevance of the Gouy phase in our experimental setup. Indeed, the Gouy phase plays a role because the eigenmodes of both the squeezed vacuum and the amplifier have rather small waists. The Gouy phase of a mode HG(n,m) is $\Phi(m,n) = (m+n+1) \arctan(z/z_R)$, where z_R is the Rayleigh length, which depends on the Gaussian mode HG(0,0) waist. From this expression, estimates show that higher-order modes like HG(2,2) exhibit Gouy phases on the order of π .

However, we believe that due to the imaging of both the squeezer modes and the pump waist on the amplifier, the Gouy phases in squeezer and amplifier modes are nearly the same. Our scheme is similar to the one where both squeezed vacuum and the pump are reflected back into the same crystal with a spherical mirror. Due to the reflection of both the pump and the squeezed vacuum, the Gouy phases of the modes in the squeezer and amplifier should be perfectly the same in the low-gain case. Because a higher gain (we have $G \sim 4.4$) changes the mode widths of the amplifier, and accordingly their Gouy phases, we increase the waist of the back-propagating pump. This should make the modes nearly overlapping, but the overlap is worse for higher-order modes, see Fig. 2(a). This effect could be responsible for the disagreement between theory and experiment for higher-order modes in Fig. 2(b).

(6) Will adding the physical description and corresponding theory of the MOPA measurement scheme at the beginning of the article better improve the article's readability?

Initially, we wanted to avoid repeating detailed descriptions of the MOPA detection scheme that were already established in previous works. Instead, we focused on emphasizing the novel aspects of our contribution, namely the real-time detection of multimode squeezing. But indeed, it is better to add some basic theory, and now we do it in the revised manuscript (lines 66-73).

(7) It can be seen from the formulas in section (6.4) of the article that the imperfect mode coupling of MOPA will reduce the measured squeezing value. In the formulas, R_{mn} represents the loss caused by imperfect mode coupling and introduces vacuum noise. However, during the MOPA process, does imperfect mode coupling only introduce vacuum noise? Are there any other amplified coupling noises? And are the formulas derived strictly?

Again, we thank the reviewer for noticing an important flaw of our consideration. Initially, the equation in Section 6.4 was written intuitively, assuming that imperfect mode coupling introduces only vacuum noise, similarly to loss. Now, we derived Eq. (6) strictly, using the mode overlap matrix. The result is that coupling with 'wrong' modes is not equivalent to loss; rather, it leads to squeezing and anti-squeezing re-distributed over several modes. In our case, the effect of this coupling on the measurement is weak, which we now show in Sec. 2.3 of the Supplementary Information and the new Fig. S6. Accordingly, it cannot explain the deviation of the experimental points from the theory, as it did before. We revised Fig.2(b) now, plotting the same experimental points with the theoretical curves accounting only for losses. The 2 to 3 dB disagreement between the experimental and theoretical values of squeezing for higher-order modes we now attribute to the difficulty in adjusting the phases for these modes through alignment. The latter relies on eliminating the interference rings in the output intensity distribution (now explained better in lines 418-423), but this procedure is not sensitive to the phases of higher-order modes.

We are very grateful to reviewer #1 for their thoughtful questions, which helped us to improve the quality of the work and even correct some errors. We hope that now the work can be published.

Reviewer #2 (Remarks to the Author):

This work addresses the long-standing challenge of real-time, loss-tolerant characterization of multimode squeezed light by proposing and experimentally validating a solution based on multimode optical parametric amplifiers (MOPAs).

We thank the reviewer for recognizing the central goal and the importance of our work.

First, spatially multimode squeezed vacuum states (with approximately 53 effective modes, and the strongest mode corresponding to ~ 9 dB of squeezing) are generated via type-I collinear degenerate parametric down-conversion (PDC) in a BiBO crystal. A MOPA is then constructed using the same BiBO crystal, and mode matching with the squeezer is achieved by optimizing the MOPA pump waist (145 μm), resulting in an overlap of over 70% for the first 8 modes. Leveraging the MOPA's ability to "map quadrature variances to output intensity," post-amplification detection becomes loss-tolerant. Subsequently, a spatial light modulator (SLM) sorts the amplified multimode light (with a sorting efficiency of $\sim 0.5\%$), and a camera real-time reads the intensities of the 9 strongest Hermite-Gaussian (HG) modes to infer their quadrature variances. Experimental results show that despite an overall detection efficiency of less than 0.3%, the fundamental mode achieves -7.9 ± 0.6 dB of pulsed squeezing (the highest value reported to date) with a quantum state purity of 63%–78% by mean intensity detection. Additionally, the system enables real-time characterization of nullifier variances and entanglement witnesses for two-node cluster states, estimates the nullifier squeezing levels of 3-to-5-node cluster states via mode contribution analysis, and proposes extending this method to frequency modes—providing a critical characterization tool for high-dimensional quantum technologies such as quantum metrology and continuous-variable quantum computing.

We thank the reviewer for the comprehensive and accurate summary of our work.

Before making a decision about the presented manuscript, the following concerns should be addressed.

1. The squeezing here is come from the mean intensity detection of the mode after amplification. It is totally different from the standard method Balanced Homodyne Detection (BHD). The proof of equivalence is necessary. At least, the authors should compare the squeezing measuring between two squeezing measuring methods for the same high mode once. I think the gain of MOPA is corresponding to a possible maximum squeezing limit.

Indeed, the reviewer is right that a reliable verification method such as balanced homodyne detection (BHD) is important. However, in our case, BHD is impossible because a suitable local oscillator at the squeezing wavelength (709.33 nm) is unavailable. This limitation actually illustrates a key strength of OPA-based detection: the same source used for squeezing generation can be employed for detection, ensuring natural compatibility, without the requirement of additional sources.

Meanwhile, squeezing detection via OPA has become a reliable method and has already been implemented in several previous works [*Nat Commun* **9**, 609 (2018), *Optica* **6**, 1233-1236 (2019), *Science* **377**,1333-1337(2022), and *Opt. Express* **28**, 34916-34926 (2020)], all of them accepted in the scientific community. Importantly, the equivalence between the OPA

scheme and BHD was recently tested (Opt. Express 28, 34916-34926 (2020)), proving the reliability of the method. Finally, our results reasonably match the expectations of the rigorous theoretical model of our multimode squeezer (Fig. 2(b) of main text).

Regarding the gain of MOPA, we have carefully analysed its effect on the inferred squeezing. We show in Supplementary Section 2.4 that the measurement systematic error due to MOPA gain is below 0.04%.

We now comment on the local oscillator issue in the revised manuscript (lines 237-241).

2. Based on the above, the expression of “the highest recorded squeezing” is inappropriate. However, it is reasonable if the authors give the proof of equivalence.

We understand the reviewer's concern regarding the equivalence between OPA-based squeezing detection and standard BHD. Due to the lack of a suitable local oscillator at 709.33 nm, such a comparison is currently not possible in our setup. Nonetheless, we recognize the value of such benchmarking and will try to address it in future work.

We also note that the prior works using the same OPA-based detection scheme have employed similar, if not stronger, phrasing, such as “few-cycle squeezing” [Science377_1333_2022], “THz-bandwidth squeezing” [NatCommun9_609_2018], and “all-optical detection” [OptExpress28_34916_2020], all of which are widely accepted in the community.

3. Could the authors provide specific details on how the pump beam is shaped? Additionally, it is a suggestion that relevant information about pump shaping be added to Fig. S1 for clarity.

As mentioned in the manuscript, this work presents a proof-of-principle demonstration of how simple spatial pump shaping, specifically through waist scaling, can influence mode matching and thereby enhance the efficiency of the detection scheme (Fig. 2). The pump waist is modified by adjusting two lenses L2 and L3 (Fig. S1).

More generally, a mode shaper such as an SLM could be used to generate arbitrary pump profiles, allowing the MOPA's modal content to be precisely engineered to match any desired set of input modes.

We now modify Fig. S1. Lines 133-134 of the manuscript explain that what we do is a first step towards arbitrary pump shaping.

4. The modes purity is also important. Output light after filter has been reduced to 0.3%, how much squeezing of remaining light is?

With OPA-based detection, all losses occurring after amplification (e.g., filtering, detection, or sorting losses) do not affect the measured squeezing or purity, since the relevant

quadrature variance has already been mapped onto mean intensity (line 56). Only losses occurring before amplification degrade the state.

The obtained purity of the detected states is shown in Fig. 2(c), which is considerably high for such amounts of squeezing.

We thank the reviewer for their detailed and constructive feedback, and hope that our responses have clarified the value of this work.

Reviewer #3 (Remarks to the Author):

We thank the co-reviewer for their contribution to the review process and appreciate their role in the collaborative evaluation. We hope that our responses satisfy their concerns.

Reviewer #4 (Remarks to the Author):

In their work titled "Real-time Monitoring of Multimode Squeezing", the authors have demonstrated an approach to simultaneously monitor multi-mode squeezed states in real time using a combination of a multimode OPA and SLM based mode-sorter. They used their approach to also characterize cluster states. The results in this work are not surprising and is expected, thanks to the loss-tolerant detection mechanism provided by OPA. The results in this paper are comprehensive. I am willing to recommend publication of this work in Nature communications, if the authors can address my concerns, that I believe will improve the clarity of the work.

We thank the reviewer for willing to recommend this work for publication. While the results may appear expected given the capabilities of OPA, this is, to our knowledge, the first experimental realization of real-time squeezing measurement across multiple co-propagating modes — a long-standing challenge in photonic quantum technologies.

1. Can the authors add more details about the mode overlap purity that can be obtained by this approach. is it possible to attain above fault-tolerant threshold in the context of CV cluster-state based quantum computing applications.

We thank the reviewer for this very insightful comment.

The answer is two-fold:

First, prior studies have shown that initial squeezing levels of 9.9 dB [PRX Quantum **3**, 010315 (2022)] and 10.5 dB [arXiv:2506.13643v1 [quant-ph]] are sufficient for achieving

fault-tolerant quantum computation, using GKP encoding. These values are not far from the pulsed squeezing we observe in the fundamental mode (-7.9 ± 0.6 dB), and with good alignment, also in several higher-order modes. We are experimentally pushing in this direction. The reviewer is also invited to check our recent work for spectral modes (arXiv:2508.04502 [quant-ph]), where we have reached homogeneous -7 dB pulsed squeezing over 60 spectral modes, potentially qualifying our source for these very purposes.

Second, with better alignment, the achievable squeezing is ultimately limited only by the source itself. We are currently investigating the limits of our source at such high-squeezing regimes. In the revised manuscript, we now comment on the potential direction of our source in fault-tolerant applications (lines 232-235).

2. What is the degradation expected with higher order modes? What is the theoretical bounds on the useful number of modes that can be addressed while still satisfying the fault-tolerant threshold given certain constraints like pump power (that can also lead to spontaneous nonlinear scattering process that can degrade the degree of squeezing), resolution of the SLM etc..

The expected purity degradation of higher order modes depends on pre-amplification losses (independent on the mode order, about 10% in our setup) and quality of alignment and mode sorting. Regarding sorting, we ensure that each measured mode contains contributions only from the targeted mode. This is achieved by selecting a very narrow region in the sorting plane (Supplementary material sec.4), although it comes at the cost of extremely high sorting losses (>99%). However, since these losses occur after amplification, they do not affect the measured squeezing or purity as long as the amplified signal is still higher than the noise.

The SLM resolution will probably restrict the number of sortable modes to several hundred, as is typically the case in current commercial devices [for instance, see the sorting of 210 spatial modes: *Nat Commun* **10**, 1865 (2019)].

Apart from the sorting, the number of useful modes depends on the modal content of the source. By focusing the pump softer or using a thinner crystal, the effective number of spatial modes can be increased up to a few hundred (line 285).

3. Can the authors comment on the scalability of the approach, especially in the context of integrated photonics platform like lithium niobate nonlinear integrated photonic circuits. How many modes can be harnessed in this platform, given the flexibility of QPM engineering, and replacing SLM with waveguide coupler based mode-sorter?

Indeed, moving toward integrated photonics is a natural direction to demonstrate our scheme, but only in the spectral domain. Waveguide-based sources significantly limit spatial

complexity, typically supporting only single- or few-mode operation. Spectrally, however, this can scale up to hundreds or thousands of modes depending on the configurations of both the pump and the crystal phase-matching.

Our source (see arXiv:2508.04502 (2025)) and experiment present a proof-of-principle concept, which, if combined with properly engineered low-loss waveguides, would substantially contribute to scalable implementations in integrated platforms.

We also note that the mode sorting stage, whether using SLMs, waveguide couplers, or nonlinear frequency conversion, would function similarly in our case, as high post-amplification losses are tolerated. Only a mode sorter after amplification is required.

In the revised manuscript, we now comment on the potential extension of our scheme to integrated photonic platforms (lines 262-265).

4. How does spatial multiplexing compare with regards to frequency-domain multiplexing with regards to real-time detection. Are there concerns of cross-talk in frequency domain approach, that is less severe in mode-division approach?

Indeed, spatial mode sorting is more direct and well-established compared to spectral-domain sorting. Nevertheless, spectral mode sorting techniques, though more complex, are actively developing. Examples include temporal interferometers (Phys. Rev. A 110, 033721 (2024)) and pulsed optical gates (PRX Quantum 4, 020306 (2023)), which enable real-time access to frequency modes. Our work, while demonstrated in the spatial domain, could in principle be extended to the spectral domain using such tools. We mentioned this in the paper (line 260-261).

We thank the reviewer for the thoughtful comments. We hope that our responses have addressed the points raised and clarified the potential and scope of our work.

List of changes (highlighted blue in the manuscript):

1. Some theory added (p.2).
2. The alignment procedure explained (lines 121-212, 419-423).
3. The spatial tailoring of the pump explained better (lines 128-134).
4. Mode overlap imperfections remaining despite the pump tailoring are discussed (lines 142-147, section 6.4, Sec. 2.3 and Fig. S6 of the SI).
5. Fig. 2(b) is modified, as well as its caption and the discussion (lines 156-157 and 161-163).
6. Fig. 3(a,b) are modified, as well as the caption and discussion in p.6. Panel (c) is added.
7. Fig. 4 is modified (phase distributions added).
8. The Discussion section is considerably extended.

9. Fig. S1 is modified, and its description.